Education about pain and experience with cognitive-based interventions do not reduce healthcare professionals’ chronic pain

http://orcid.org/0000-0002-4174-7149 Weisman Asaf 1 asafweisman@gmail.com
http://orcid.org/0000-0003-4809-490X Yona Tomer 2
Masharawi Youssef 1
1 Spinal Research Laboratory, Department of Physical Therapy, Stanley Steyer School of Health Professions, Gray Faculty of Medical and Health Sciences, Tel-Aviv University , Tel-Aviv , Israel
2 Department of Biomedical Engineering, Technion-Israel Institute of Technology , Haifa , Israel
Chen Yung-Sheng
Electronic publication date: 2025 May 30
Publication date: 2025
Volume: 13
Electronic Location ID: e19448
Received 2024 Oct 17; Accepted 2025 Apr 20
Copyright: © 2025 Weisman et al.
Copyright year: 2025
Copyright holder: Weisman et al.
License: This is an open access article distributed under the terms of the Creative Commons Attribution License, which permits unrestricted use, distribution, reproduction and adaptation in any medium and for any purpose provided that it is properly attributed. For attribution, the original author(s), title, publication source (PeerJ) and either DOI or URL of the article must be cited.
License URL: https://creativecommons.org/licenses/by/4.0/

Keywords: Pain neuroscience education, Cognitive behavioral therapy, Acceptance commitment therapy, Mindfulness, Chronic pain

Funding: The authors received no funding for this work.

==============================
Background

Cognitive-based interventions like pain neuroscience education (PNE), cognitive behavioral therapy (CBT), acceptance commitment therapy (ACT), and mindfulness meditation are popular for managing chronic pain. Despite their widespread adoption, evidence for their efficacy remains contradictory. Healthcare professionals (HCPs) represent a unique population to evaluate these approaches, as they possess specialized knowledge about pain mechanisms and often implement these interventions with patients. The logical premise underlying cognitive-based interventions suggests that increased knowledge and cognitive engagement with pain concepts should reduce pain intensity, making educated HCPs with chronic pain an ideal test case for this theoretical framework.

Purpose

To investigate whether HCPs with chronic pain who (HCPs+CP) are familiar with these methods experience less pain and improved quality of life compared to less experienced HCPs+CP and healthy HCPs (H-HCPs).

Methods

This cross-sectional study used an anonymous online questionnaire distributed in English through closed professional social media groups internationally. Data were collected from 550 HCPs (319 healthy, 231 with chronic pain) primarily from Israel, Canada, United States, United Kingdom, and Australia. Participants were categorized by their knowledge of pain neuroscience, experience with cognitive-based interventions, and chronic pain type (primary or secondary). Pain intensity was measured using the Numerical Pain Rating Scale, and quality of life was assessed with the World Health Organization tool the WHOQOL-BREF. Statistical analyses included Spearman’s correlation tests and independent samples t-tests.

Results

Pain intensity did not significantly differ between primary pain group (4.24 ± 2.21) and secondary pain group (4.37 ± 2.33) (t = −0.267, p = 0.79, Cohen’s d = −0.05). The total WHOQOL score was lower in HCPs+CP (66.81 ± 15.74) compared to healthy HCPs (71.13 ± 14.02) (t = −2.136, p = 0.035), but after removing respondents unfamiliar with interventions, no significant differences remained between groups across all WHOQOL domains (−0.09 < Cohen’s d < 0.14, p > 0.05). Among the 146 H-HCPs who had recovered from chronic pain, only 11% attributed their recovery to cognitive-based interventions, while the majority credited physical therapy (37.7%) and spontaneous recovery (32.9%).

Conclusion

Despite their specialized knowledge and experience with cognitive-based interventions, HCPs+CP did not report reduced pain intensity, though they maintained quality of life comparable to healthy colleagues. These findings challenge current theoretical models underlying cognitive-based pain management.

Introduction

Pain is the most common reason that leads people to seek medical advice (Cohen, Vase & Hooten, 2021). Moreover, persistent (chronic) pain is a growing global health problem that poses significant personal, societal, and economic burdens (Cohen, Vase & Hooten, 2021; Mills, Nicolson & Smith, 2019). Although understanding the neurobiology of chronic pain states has advanced considerably, there has been little progress in their treatment (Schug & Goddard, 2014).

Pain is prevalent among healthcare professionals (HCPs), affecting individuals across a range of specialties (Ali et al., 2024; Shaygan et al., 2022; Weisman et al., 2021; Yona et al., 2022). Chronic pain is a significant occupational health concern for these workers, with prevalence estimates ranging from 30% to over 89% across various specialties and regions, and low back pain is the most commonly reported issue (Ali et al., 2024; Shaygan et al., 2022; Yamada et al., 2016). Moreover, higher prevalence rates are observed among women, younger professionals, and rotating shift workers (Ali et al., 2024; Shaygan et al., 2022; Yamada et al., 2016). However, most epidemiological data have been derived from a limited number of medical professions, highlighting a need for further research on HCPs.

In the search for novel and effective interventions for persistent pain, some HCPs have looked towards non-pharmaceutical interventions, such as pain neuroscience education (PNE), cognitive behavioral therapy (CBT), acceptance commitment therapy (ACT), and mindfulness meditation which have become increasingly popular (Ehde, Dillworth & Turner, 2014; Weisman et al., 2022). PNE is a group of educational programs with the premise that education based on current concepts and opinions from pain neuroscience can contribute to positive outcomes for participants, such as an enhanced ability to function and lessening fear and anxiety (Moseley et al., 2024; Weisman et al., 2022). CBT principles suggest that psychological problems arise from flawed thinking and maladaptive behavior patterns and can be overcome through learning and adopting more effective coping strategies, leading to symptom relief and improved functioning (Fenn & Byrne, 2013). ACT is another behavioral and cognitive intervention that employs acceptance, mindfulness meditation, commitment, and behavior change processes to foster psychological flexibility, enabling individuals to engage more consciously and flexibly with their psychological experiences and enhance values-based action (Hayes et al., 2013).

The rationale for these interventions can be traced to attempts to apply the “biopsychosocial model of pain” (Nicholas, 2022). According to this model, the experience of pain is causally composed of biological, psychological, and sociological factors. Therefore, in theory, an informed clinician should be able to identify the likely cause(s) (Nicholas, 2022). This reasoning is often used to explain the use of psychological, educational, and cognitive interventions. Another rationale for using these interventions derives from constructivist and active inference models that view the experience of pain results in some unexplained way from the brain’s “prediction about bodily harm, shaped by sensory input and context-based predictions” (Ashar et al., 2022). Accordingly, the mere fear of tissue damage may lead some individuals to experience innocuous somatosensory input as painful (Ashar et al., 2022). It logically follows that interventions that change a patient’s context-based predictions can favorably change their experience of pain.

A specific context that appeals to the proponents of these interventions is chronic pain, where no identifiable tissue damage or disease processes can be identified. Such pain states are coded in the ICD-11 as chronic primary pain (Nicholas et al., 2019). Those promoting these interventions for primary chronic pain assert that in the absence of identifiable tissue damage or disease process, the pain is likely to be of “psychological origin” and should, therefore, be more responsive to educational and cognitive-based interventions than would secondary chronic pain (Ashar et al., 2022; Donnino et al., 2021).

The guiding principle here is that psychological interventions can be central to pain reduction, which has led us to posit that the most suitable chronic pain population to assess whether or not those methods might be valuable is informed HCPs experiencing chronic pain, constituting a unique population (Weisman et al., 2022). In our view, some HCP groups, like physiotherapists, can be considered to be the most educated clinicians in pain theory and practice. In North America, it has been reported that medical students receive an average of 5 h of pain education. In contrast, American physiotherapists receive 31 h and Canadians 24.9 (Hoeger Bement & Sluka, 2015; Watt-Watson et al., 2009).

According to the mentioned theories, we hypothesized that chronic pain intensity would negatively correlate with varying levels of pain education among HCPs experiencing chronic pain. We also assumed that well-informed HCPs with chronic pain would be more likely to seek solutions for their pain. It seems plausible that these HCPs, frequently treating individuals experiencing chronic pain and utilizing cognitive-based interventions, would recognize personal psychological traits that may impede progress in pain rehabilitation. Given that HCP proponents of these theories endorse these interventions as tools for chronic pain management, it seems reasonable to expect that their own pain levels and quality of life might reflect adherence to the strategies they advocate for their patients, providing a practical, real-world “proof of concept” of the interventions’ efficacy.

We used the following formal logical argument to justify our rationale for performing the study as well as our choice of HCPs living with chronic pain (Box 1):

Box 1 Formal logical argument for study rationale and hypothesis.

*It is possible to place the complete argument into a large language model like ChatGPT or Claude to assess the soundness and validity of the argument.

Box 1: Formal Logical Argument for Study Rationale and Hypothesis Let:

C = Cognitive activities (thinking, reasoning, remembering, imagining,

learning, using language).

L = Learning (gaining knowledge or understanding through study,

instruction, or experience).

I = Cognitive-based interventions.

P = Pain intensity reduction.

Q = Quality of life improvement.

H = Healthcare professionals (HCPs).

K = Knowledge about pain and cognitive-based interventions

R = Reduced pain.

Premises:

P1: ∀x(Cx → (Lx ∧ Sx)) [For all x, if x is a cognitive activity, then x

involves learning and study]

P2: ∀x(Ix → (Px ∧ Qx)) [For all x, if x is a cognitive-based intervention,

then x reduces pain intensity and improves quality of life]

P3: Most x (Hx → (Kx ∧ ∃y(Iy ∧ Kxy))) [For most x, if x is an HCP, then x has knowledge about pain and there exists a cognitive-based intervention y that x is knowledgeable about].

Conclusion:

C: Most x∀y((Hx ∧ CPx ∧ Kxy ∧ Iy) → Rx) [For most x and y, if x is an

HCP with chronic pain and is knowledgeable about cognitive-based

intervention y, then x experiences reduced pain].

Premise 1, “Cognitive” definition is: “of, relating to, or being conscious mental activities (as thinking, reasoning, remembering, imagining, learning words, and using language)” (merriam-webster dictionary). This definition also pertains to “learning,” where to “learn” is “to gain knowledge or understanding of or skill in by study, instruction, or experience”, and “study” is “application of the mental faculties to the acquisition of knowledge” (Merriam-Webster dictionary).

Premise 2: Cognitive-based interventions (those that utilize “cognition”) reduce pain intensity and improve quality of life.

Premise 3: Many HCPs are highly educated about pain and possess above average knowledge in this area, with many also being well-versed in cognitive-based interventions (as outlined in premise 1).

Conclusion: Therefore, HCPs with chronic pain (HCPs+CP) who are knowledgeable about cognitive-based interventions should experience less pain (P) and improved quality of life (Q) compared to HCPs+CP who are less familiar with these interventions.

Objectives: The primary objective of this exploratory study is to assess the relationship between healthcare professionals’ familiarity with cognitive-based interventions and their chronic pain outcomes. Specifically, we aim to determine whether increased knowledge and experience with interventions such as pain neuroscience education (PNE), cognitive behavioral therapy (CBT), and related approaches are associated with lower pain intensity and improved quality of life among healthcare providers. To address this objective, we designed the study to examine the following hypotheses: (1) HCPs experiencing “chronic primary pain” (PrimaryPG according to ICD-11) and are familiar with PNE and experienced with cognitive-based interventions will demonstrate significant negative correlations between pain intensity, familiarity, and experience compared to HCPs with “chronic secondary pain” (SecondaryPG according to ICD-11). Additionally, HCPs from the PrimaryPG will have significantly lower pain intensity and better quality of life compared to HCPs from the SecondaryPG (Treede et al., 2019) with the same level of familiarity and experience with these interventions (one-tailed hypothesis).

(2) HCPs+CP who experience chronic pain (and are familiar with the educational content of PNE (education/familiarity variable) and use cognitive-based interventions regularly (experience variable) will report lower pain intensities (dependent variable) compared to HCPs+CP who are less familiar with and less experienced in these interventions (one-tailed hypothesis). We expected to observe negative correlations between pain intensity, familiarity, and experience variables (i.e., the more educated/familiar and experienced HCPs are with these interventions, the lesser their reported pain intensity should be).

(3) The quality of life of HCPs+CP, who are familiar with PNE and are experienced with cognitive-based interventions, should be comparable in these parameters to healthy HCPs (H-HCPs) (one-tailed hypothesis).

(4) H-HCPs whose pain has resolved will perceive this resolution as attributable to educational and/or cognitive-based interventions.

Methods

Study design

Observational, cross-sectional.

Human ethics

Tel-Aviv University’s Ethical Review Board approved this study before data collection (Date: 06.06.2022, Number: 0005039-2). All the study’s procedures were conducted according to Israel’s Ministry of Health regulations and the Good Clinical Practice Guidelines (GCP).

Study tools

Findings are reported according to the “Strengthening the Reporting of Observational Studies in Epidemiology” (STROBE) statement (Little et al., 2009). Alchemer (https://www.alchemer.com/) was used as the platform for the international distribution of the anonymous questionnaire in English.

Questionnaire development and validation

Before designing the questionnaire, the authors conducted a thorough review of the relevant literature to identify key variables related to pain intensity, quality of life, and familiarity with cognitive-based interventions. Based on this review, we collaboratively selected the most relevant items to capture these constructs effectively. A Likert scale format was chosen for many of the questionnaire items due to its ease of use, ability to measure varying degrees of agreement or experience, and demonstrated reliability in assessing subjective perceptions in pain research (Norman, 2010; Sullivan & Artino, 2013). In addition, Likert scales facilitate statistical analysis because their ordinal properties allow for a wide range of parametric and non-parametric tests, making them a popular choice in survey-based research (Boone & Boone, 2012; Carifio & Perla, 2007).

Given the exploratory nature of our study and its primary aim of hypothesis generation and testing rather than clinical application, we did not conduct formal pilot testing or reliability studies for the questionnaire. Our approach prioritized minimizing respondent burden and ensuring a broad data collection, which is common in early-phase exploratory research. Previous literature has suggested that extensive instrument validation may be deferred until preliminary hypotheses have been generated and the construct under investigation is better understood (Leon, Davis & Kraemer, 2011). Additionally, Rolstad, Adler & Rydén (2011) emphasize that in exploratory studies, the trade-off between instrument length and response burden is critical, and that extensive validation efforts are more appropriate for studies intended to inform clinical decision-making. We decided this approach will allow us to rapidly gather data while still capturing the essential constructs related to pain intensity, quality of life, and familiarity with cognitive-based interventions.

Inclusion and exclusion criteria and informed consent

We invited members of allied healthcare professions to join the study, mainly through advertisements in closed international social media groups in the English language. This method has been found to be valid to reach the desired target populations in online surveys (Weisman et al., 2023, 2021; Yona et al., 2020). On the landing page, participants were first informed that the study is intended only for HCPs such as medical doctors, physiotherapists, occupational therapists, nurses, psychologists, osteopaths, chiropractors, psychologists, trainers, bodyworkers, massage therapists and that pressing the “next” button implies their informed consent. Informed consent was obtained from all respondents; those under 18 were not allowed to participate in the study. Before beginning the study, we decided that incomplete questionnaires would be excluded from the final analysis to ensure data integrity and minimize missing data issues. Participants did not receive any monetary or material incentives for completing the survey, nor were they given the option to receive a copy of their responses.

Sample size determination

The desired sample was calculated a priori with GPower 3.1.9.4. We used an exact test family for normal bivariate correlations to test hypotheses 1 and 2 (discovering effects on pain intensity). We used the following input parameters for a stricter two-tailed hypothesis: a minimum detectable effect size of ρ = 0.2, α = 0.05, and β = 0.85; the desired sample was n = 221 for the HCPs group with chronic pain.

Study flow, funneling, and grouping

The following sections depict the study flow and grouping we employed to achieve our aims. The study flow is also available in Fig. 1.

Figure 1 Study flow and funneling.

H-HCPs, Healthy Healthcare Professionals Group; HCPs+CP, Healthcare Professionals with Chronic Pain Group; PrimaryPG, Primary Pain Group; SecondaryPG, Secondary Pain Group; WHO, World Health Organization.

Demographics

The questionnaire included basic demographics, including country, age, gender, weight, height, physical activity, alcohol consumption, smoking (yes/no and the number of cigarettes per day), education, clinical experience in years, weekly work hours, and the extent of experience treating chronic pain patients.

Pain profiling and grouping for hypotheses testing

To evaluate hypotheses 1 and 2, we employed a stratified approach based on the etiology of chronic pain among participants. We initiated this stratification by administering binary yes/no questions to ascertain a history of significant medical interventions or conditions like cancer, chemotherapy, irradiation therapy, autoimmune diseases, and recent orthopedic trauma or surgery. Participants affirming chronic pain and any of these conditions were categorized under Secondary Pain Group (SecondaryPG), presuming their pain likely stemmed from these identifiable medical conditions. Furthermore, we inquired whether participants had received specific diagnostic labels associated with chronic pain conditions, such as endometriosis or neuropathy. Receipt of such diagnoses also led to classification within the SecondaryPG. Conversely, participants who reported chronic pain but denied any of the specified medical conditions or confirmed they had never been given any official diagnoses or labels for their conditions were classified under the Primary Pain Group (PrimaryPG), assuming their pain to be non-specific (Nicholas et al., 2019).

To further test hypotheses 1 and 3, we performed grouping according to the yes/no question: “Are you currently experiencing pain lasting more than 3 months (i.e., persistent/chronic pain)?” This question dichotomized the sample into the H-HCPs and HCPs+CP.

To test hypothesis 4 and also to assess for survivorship bias (see bias reduction strategies section), only the H-HCPs who answered that they currently have no chronic pain received the following yes/no question: “Have you had chronic pain in the past that has disappeared spontaneously or due to an intervention?” Respondents who answered yes to this question received the following question: “What intervention resolved your pain (you may choose up to 3 answers)?” which included the following possible answers: Nothing-spontaneous recovery, movement/exercise, rehabilitation, pharmaceutical, educational/cognitive interventions, surgical, alternative medicine, and other. We used the term ‘resolved’ in that part of the questionnaire and also throughout the manuscript when referring to pain cessation, as it more neutrally describes the disappearance of pain without implying specific mechanisms or patient agency.

Familiarity/education and experience with cognitive-based interventions

This section followed the pain profiling section. To test the first three hypotheses with the variables of familiarity and experience with cognitive-based interventions, we used ordinal Likert scale questions (4–5 options max) (Table 1).

Table 1 Education/familiarity and experience in using cognitive-based interventions questionnaire.

A) How familiar are you with educational, therapeutic approaches for explaining pain? Examples: Pain Neuroscience Education (PNE), Explain Pain (EP), or Pain Reprocessing Therapy (PRT)? (1) Not at all.

(2) A little.

(3) Moderately familiar (heard and read about it).

(4) Very much familiar (attended at least one course or workshop).

(5) Extremely familiar (attended more than two courses and actively seeking to be updated with this content).

	G) How familiar are you with Mindfulness? (1) Not at all.

(2) A little.

(3) Moderately familiar (heard and read about it).

(4) Very much familiar (attended at least one course or workshop).

(5) Extremely familiar (attended more than two courses and actively seeking to be updated with this content).

	
B) How often would you say you use the educational concepts of explaining pain in your daily work with your patients? (1) Not at all.

(2) On rare occasions.

(3) A moderate amount (About a patient per week).

(4) Very much (at least a few patients per week).

(5) On a regular basis.

	H) How often would you say you use Mindfulness in your daily work with your patients? (1) Not at all.

(2) On rare occasions.

(3) A moderate amount (About a patient per week).

(4) Very much (at least a few patients per week).

(5) On a regular basis.

	
C) Please enter an estimated number of years using explaining pain or other PNE approaches in your practice: (Enter a number)	I) Please enter an estimated number of years using explaining pain or other PNE approaches in your practice: (Enter a number)	
D) How familiar are you with cognitive therapeutic approaches such as Cognitive Behavioral Therapy (CBT) and Acceptance Commitment Therapy (ACT)? (1) Not at all

(2) A little (heard about it).

(3) Moderately familiar (heard and read about it).

(4) Very much familiar (attended at least one course or workshop).

(5) Extremely familiar (attended more than two courses and actively seeking to be updated with this content).

	J) Do you practice mindfulness yourself? (1) Yes.

(2) No.

	
E) How often would you say you use CBT and ACT in your day-to-day work with your patients? (1) Not at all.

(2) On rare occasions.

(3) A moderate amount (About a patient per week).

(4) Very much (at least a few patients per week).

(5) On a regular basis.

	K) How often do you practice mindfulness? (1) Once a week.

(2) 2–3 times a week.

(3) 4–5 times a week.

(4) Every day.

	
F) Please enter an estimated number of years using CBT or ACT in your practice: (Enter number)		

Numerical pain rating scale

The Numerical Pain Rating Scale (NPRS) is a widely used unidimensional tool for assessing pain intensity. Respondents are asked to rate their pain on a scale from 0 (no pain) to 10 (worst imaginable pain) (Hawker et al., 2011). Its simplicity and ease of administration make it valuable in both clinical practice and research, and its strong reliability and validity have been demonstrated across various patient populations. Moreover, the NPRS has been shown to yield reliable and valid measurements when administered electronically, supporting its use in online surveys and web-based research (Hjermstad et al., 2011).

Quality of life

The last section of the study included the World Health Organization’s Quality of Life Questionnaire (WHOQOL), an established tool with stable psychometrics (Skevington, Lotfy & O’Connell, 2004). All the respondents in the study filled out the questionnaire. We also created a new variable for the calculated four WHOQOL domain scores according to the WHO’s instructions (World Health Organization, 1996). This variable includes an aggregated result of all the domains (“WHOQOL total” variable). The WHOQOL is scored by calculating mean scores for each domain, which are then transformed to a 0–100 scale. Higher scores indicate better quality of life.

Bias reduction strategies

To avoid hypothesis guessing, we introduced the study in the advertisements and landing page as an epidemiological survey of pain among HCPs. Nowhere have we mentioned it has anything to do with familiarity and experience in using educational and cognitive-based interventions. Following the demographics section, we first introduced pain profiling questions to prevent bias in reporting the familiarity and experience variables. We introduced the specific cognitive-based intervention questions only after the respondents finished their pain profile section. We disabled the option to navigate backward in the questionnaire to prevent changing the pain profile answers once the respondents were exposed to the familiarity and experience variables.

To avoid recall bias in the pain intensity reporting, we asked for an estimate of their average pain intensity in the last week using the NPRS and avoided questions about pain intensity in the long term.

A specific bias we anticipated and proactively aimed to reduce is survivorship bias. We speculated that the group of H-HCPs might include individuals who previously responded to educational and cognitive interventions. Therefore, we added a specific question to that group which is threaded with Hypothesis 4 of this study (see previous “Pain Profiling and Grouping” section). This approach allowed for an assessment of the perceived effectiveness of various interventions among HCPs who had experienced recovery from chronic pain while mitigating the impact of survivorship bias in the study’s findings.

The study was disseminated and advertised mainly through closed social media groups for HCPs on Reddit, Facebook, and the authors’ Twitter accounts. This dissemination strategy was found to be valid in reaching the targeted population (Weisman et al., 2021, 2022; Yona et al., 2022). Multiple entries were addressed by tracking cookies and IP addresses of computers and cell phones. The study was live in the first 2 weeks of August 2022 and terminated once the desired sample was slightly exceeded. Such a short dissemination period is most likely insufficient to facilitate the development and deployment of automated response systems by potential malicious actors. Notwithstanding, we employed a multifaceted approach to detect potential automated responses, encompassing continuous surveillance of response latencies for recurring temporal patterns, monitoring of IP addresses for suspicious submission frequencies, and geospatial analysis of respondent locations to identify anomalous distribution patterns that deviated from expected survey participation regions.

Statistical analysis

Data were analyzed using IBM SPSS Statistics for Windows, version 25.0 (IBM Corp., Armonk, NY, USA). We set the significance level a priori at α = 0.05. To determine whether to use parametric or non-parametric tests, normality was assessed using the Kolmogorov-Smirnov test and visual inspection of the data through boxplots (Ghasemi & Zahediasl, 2012). For hypothesis 1: A significant difference with higher pain scores for the SecondaryPG was expected to be observed. Independent samples t-test was used. Spearman’s tests were performed to assess correlations between pain intensity and familiarity and experience for this hypothesis.

For hypothesis 2: Negative correlations between pain intensity, familiarity/education, and experience variables in the HCPs+CP group were expected. Spearman’s correlation tests were used.

For hypothesis 3: A similarity or slight significance between H-HCPs and the HCPs+CP group was expected to be observed. Independent samples t-test was used.

For hypothesis 4: It was expected that the perceptions of HCPs should be reflected in the proportions of the answers to the question “What intervention resolved your pain (you may choose up to 3 answers)?” It was expected that the proportions of those who chose education and cognitive-based interventions would be equal or greater than those who chose other answers.

For the main outcomes of pain intensity and quality of life, effect sizes were calculated using Cohen’s d. Effect sizes were considered negligible for values 0.0–0.2, small for 0.2–0.5, medium for 0.5–0.8, and large for values greater than 0.8 (Cohen, 2013). Incomplete questionnaires were excluded from the final analysis to avoid missing data and imputation. For other non-hypothesis-related variables, differences between groups were determined using t-tests on continuous variables, and the distribution of nominal variables was done using the chi-square test. For multiple correlations of ordinal and interval variables or a combination of continuous and ordinal variables, we used Spearman’s ranks test with Bonferroni correction. The Bonferroni correction was calculated by dividing the original alpha level (0.05) by the total number of comparisons performed (21), resulting in an adjusted significance threshold of p < 0.00238 (0.05/21 ≈ 0.00238). This more stringent criterion was used to determine the statistical significance of all correlations. We considered correlation values of r 0–0.1 as negligible, 0.1–0.39 as weak, 0.40–0.69 as moderate, 0.7–0.89 as strong, and 0.9–1 as very strong correlation (Schober, Boer & Schwarte, 2018).

Results

General descriptives

A total of 861 respondents entered the survey, and 550 completed the entire survey (63.8%) and were included in the final analysis. The average completion time was 7.03 ± 5.5 min. Overall, there were 202 men and 348 women respondents. The top 5 country entries were from Israel (28.7%), Canada (22.8%), the United States of America (15%), the United Kingdom (13.1%), and Australia (5.7%). The proportions of the various professions were as follows: physical therapists (44%), massage therapists (21.1%), chiropractors (7.2%), osteopaths (7.1%), other therapists (6.5%), medical doctors (3.2%), nurses (3.2%), occupational therapists (2.6%), bodyworkers (2.2%), fitness instructor/trainer (2%) and psychologists (1%).

Only 107/231 HCPs with chronic pain received diagnostic labels. The labels included in the SecondaryPG were endometriosis (n = 5) and neuropathy (n = 25). In the PrimaryPG, they were Chronic Fatigue Syndrome (n = 4), Complex Regional Pain Syndrome (n = 2), Ehlers-Danlos Syndrome (n = 1), headache disorders (n = 11), migraines (n = 27), Irritable Bowel Syndrome (n = 19), Post Traumatic Stress Disorder (n = 6), trigeminal neuralgia (n = 4), and vulvodynia (n = 3).

HCPs without chronic pain (n = 319) predominantly consisted of physical therapists (50.7%), massage therapists (15.9%), and chiropractors (8.7%). They reported high levels of physical activity, with 85% exercising at least 2–3 times per week. The majority (75%) consumed alcohol, with 82% drinking once per week or less. Their educational background was strong, with 92.5% holding at least a bachelor’s degree. These professionals primarily worked in the private sector (49.5%) or public sector (30%), with 20.5% working in both. They frequently treated chronic pain patients, with 77.5% treating them very often or daily. While most were familiar with pain neuroscience education (PNE) and mindfulness (88% and 92.5% reporting at least some familiarity, respectively), they showed lower familiarity with CBT/ACT techniques. Approximately half (53%) practiced mindfulness themselves, with 66.7% of those practitioners doing so at least 2–3 times per week.

Comparison and exploration of PrimaryPG to SecondaryPG (hypothesis 1)

The demographics of the continuous, nominal, and ordinal variables of the PrimaryPG (n = 174) and SecondaryPG (n = 57) were first compared. The data indicated that the groups were similar, with no significant differences in all the collected variables (Tables 2 and 3). The average pain intensity in the past week showed no significant difference between groups (t = −0.267, p = 0.79). The effect size for this comparison was negligible (Cohen’s d = −0.05), indicating minimal practical difference in pain intensity between groups. Similarly, effect sizes for all WHOQOL domain comparisons between primary and secondary pain groups were negligible (−0.17 < Cohen’s d < 0.03), further confirming minimal practical differences in quality-of-life measures. Exploration of correlations of the dependent variable (pain intensity) with all other independent variables (familiarity, experience, and WHOQOL) indicated that the only significant correlations were in the PrimaryPG with the WHOQOL psychological, social, environmental domains and its total score (−0.259 < rs < −0.196, 0.003 < p < 0.028). Following the Bonferroni correction, none of the correlations remained statistically significant at the adjusted alpha level of 0.05 (Table 4).

Table 2 Continuous and ordinal demographics of healthcare professionals experiencing chronic pain.

Variables	PrimaryPG
(n = 174)	SecondaryPG
(n = 57)	Significance	
Gender			χ2 = 3.745, p = 0.053	
Men (%)	56 (32%)	9 (16%)	
Women (%)	118 (68%)	48 (84%)	
Age	43.59 ± 11.04	42.47 ± 11.33	t = 0.657, p = 0.512	
Weight (kg)	76.01 ± 17.83	79.72 ± 18.43	t = −1.349, p = 0.179	
Height (cm)	169.98 ± 11.42	169.47 ± 9.07	t = −0.304, p = 0.762	
BMI	26.30 ± 5.73	27.94 ± 6.60	t = −1.78, p = 0.076	
Average pain intensity in the past week (NPRS)	3.79 ± 1.71	3.88 ± 1.754	t = −0.267, p = 0.79
Cohen’s d = −0.05	
Years of living with pain	8.26 ± 9.88	9.84 ± 10.55	t = −0.816, p = 0.416	
Number of smokers (%)	n = 11(6%)	n = 1(1.7%)	χ2 = 1.819, p = 0.177	
Number of cigarettes per day	8.21 ± 9.15	5 ± 0.00	t = 0.339, p = 0.740	
Clinical experience (years)	15.95 ± 10.72	14.47 ± 10.74	t = 0.904, p = 0.367	
Weekly work hours	33.10 ± 13.49	31.6 ± 10.96	t = 0.764, p = 0.445	
Years of using PNE	7.03 ± 6.957	6.61 ± 6.781	t = 0.652, p = 0.726	
Years of using CBT, ACT	5.44 ± 5.672	3.87 ± 3.635	t = 1.763, p = 0.082	
Years using mindfulness	6.03 ± 6.42	6.06 ± 6.621	t = -0.030, p = 0.976	
WHOQOL
Physical domain	13.08 ± 1.57	13.27 ± 1.60	t = −0.758, p = 0.449
Cohen’s d = −0.12	
WHOQOL
Psychological domain	13.90 ± 1.84	13.84±1.84	t = 0.199, p = 0.842
Cohen’s d = 0.03	
WHOQOL
Social domain	13.96 ± 3.28	14.50 ± 2.48	t = −1.142, p = 0.254
Cohen’s d = −0.17	
WHOQOL
Environmental domain	15.41 ± 2.29	15.74 ± 2.20	t = −0.966, p = 0.335
Cohen’s d = −0.15	
WHOQOL
Total score	56.35 ± 7.08	57.35 ± 6.16	t = −0.956, p = 0.340
Cohen’s d = –0.15	
Note:

PrimaryPG, Primary Pain Group; SecondaryPG, Secondary Pain Group; BMI, Body Mass Index; NPRS, Numerical Pain Rating Scale; PNE, Pain Neuroscience Education; CBT, Cognitive Behavioural Therapy; ACT, Acceptance Commitment Therapy; WHOQOL, World Health Organization’s Quality of Life Questionnaire.

Table 3 Proportions of ordinal and nominal demographics of healthcare professional experiencing chronic pain.

Variables	PrimaryPG
(n = 163)	SecondaryPG
(n = 68)	Significance	
Profession	Bodyworker = 1.7%

Chiropractor = 5.7%

Fitness Instructor/Trainer = 1.7%

Massage therapist = 30%

Medical doctor = 1.7%

Nurse = 4.5%

Occupational Therapist = 3.5%

Osteopaths = 5%

Physical Therapist = 36%

Psychologist = 1.7%

Other = 8.5%

	Bodyworker = 1.7%

Chiropractor = 3.5%

Fitness Instructor/Trainer = 0%

Massage therapist = 36%

Medical doctor = 0%

Nurse = 5.2%

Occupational Therapist = 0%

Osteopaths = 12.2%

Physical Therapist = 33.3%

Psychologist = 0%

Other = 8.5%

	χ2 = 9.169, p = 0.516	
Physical activity	Not at all = 3%

A little (once a week) = 14%

Moderately (2–3 times a week) = 40%

Very much (3–4 times a week = 27%

Extremely (5–7 times a week) = 16%

	Not at all = 6%

A little (once a week) = 9%

Moderately (2–3 times a week) = 39%

Very much (3–4 times a week = 28%

Extremely (5–7 times a week) = 18%

	χ2 = 5.680, p = 0.224	
Alcohol consumption	Not at all = 28%

A little (once per month) = 32%

Moderately (1 time per week) = 23%

Very much (2–3 times a week) = 14%

Extremely (4–7 times a week) = 3%

	Not at all = 25%

A little (once per month) = 28%

Moderately (1 time per week) = 29%

Very much (2–3 times a week) = 14%

Extremely (4–7 times a week) = 4%

	χ2 = 4.827, p = 0.305	
Education	Less than 12 years = 2%

High school diploma = 10%

Academic Bachelors = 46%

Academic Masters = 29%

Academic > Masters = 13%

	Less than 12 years = 3%

High school diploma = 13%

Academic Bachelors = 45%

Academic Masters = 25%

Academic > Masters = 14%

	χ2 = 0.754, p = 0.944	
Sector	Public = 34%

Private = 50%

Both = 16%

	Public = 33%

Private = 50%

Both = 17%

	χ2 = 0.013, p = 0.993	
How often you treat patients with chronic pain	Never = 1%

Sometimes = 15%

Very often = 53.5%

Everyday = 30.5%

	Never = 2%

Sometimes = 19%

Very often = 33.3%

Everyday = 45.7%

	χ2 = 7.238, p = 0.065	
Familiarity with PNE	Not at all = 28%

A little (once per month) = 32%

Moderately (1 time per week) = 23%

Very much (2–3 times a week) = 14%

Extremely (4–7 times a week) = 3%

	Not at all = 28%

A little (once per month) = 32%

Moderately (1 time per week) = 23%

Very much (2–3 times a week) = 14%

Extremely (4–7 times a week) = 3%

	χ2 = 7.318, p = 0.120	
How often you use PNE?	Not at all = 8%

On rare occasions = 18.6%

A moderate amount = 32%

Very much = 21.3%

On regular basis = 20.1%

	Not at all = 10%

On rare occasions = 14%

A moderate amount = 30%

Very much = 20

On regular basis = 26%

	χ2 = 1.552, p = 0.817	
Familiarity with CBT, ACT	Not at all = 9.7%

A little = 21.8%

Moderately familiar = 39%

Very familiar = 22.9%

Extremely familiar = 6.6%

	Not at all = 14%

A little = 21%

Moderately familiar = 39.5%

Very familiar = 19.2%

Extremely familiar = 6.3%

	χ2 = 1.125, p = 0.890	
How often you use CBT, ACT	Not at all = 39%

On rare occasions = 25%

A moderate amount = 18.5%

Very much = 11.5%

On regular basis = 6%

	Not at all = 32%

On rare occasions = 43%

A moderate amount = 6%

Very much = 8.5%

On regular basis = 10.5%

	χ2 = 9.347, p = 0.053	
Familiarity with mindfulness	Not at all = 7.5%

A little = 23%

Moderately familiar = 31%

Very familiar = 26.5%

Extremely familiar = 12%

	Not at all = 3.5%

A little = 17.5%

Moderately familiar = 35%

Very familiar = 28%

Extremely familiar = 16%

	χ2 = 2.347, p = 0.672	
How often you use mindfulness	Not at all = 7.5%

On rare occasions = 26.5%

A moderate amount = 23%

Very much = 23.5%

On regular basis=19.5%

	Not at all = 3.5%

On rare occasions = 24.5%

A moderate amount = 24.5%

Very much = 19%

On regular basis = 28.5%

	χ2 = 5.136, p = 0.400	
Do you practice mindfulness yourself?	Yes = 60%

No = 40%

	Yes = 56%

No = 44%

	χ2 = 0.180, p = 0.671	
How often do you practice mindfulness	Once a week = 33%

2–3 times a week = 30%

4–5 times a week = 9%

Everyday = 25%

	Once a week = 29%

2–3 times a week = 13%

4–5 times a week = 21

Everyday = 37%

	χ2 = 6.599, p = 0.086	
Note:

PrimaryPG, Primary Pain Group; SecondaryPG, Secondary Pain Group; PNE, Pain Neuroscience Education; CBT, Cognitive Behavioural Therapy; ACT, Acceptance Commitment Therapy.

Table 4 Spearman’s correlations of pain intensity in the last week with familiarity/education, experience, and quality of life variables.

Variables	PrimaryPG (n = 174)	SecondaryPG (n = 57)	Pooled together as HCPs+CP (n = 231)	Adjusted p-value	
Familiarity with PNE	rs = −0.135, p = 0.130	rs = −0.104, p = 0.559	rs = −0.138, p = 0.082	p = 1.000	
How often do you use PNE	rs = −0.064, p = 0.516	rs = 0.039, p = 0.847	rs = −0.42, p = 0.633	p = 1.000	
Years using PNE	rs = −0.092, p = 0.368	rs = 0.235, p = 0.281	rs = −0.024, p = 0.791	p = 1.000	
Familiarity with CBT and ACT	rs = −0.051, p = 0.535	rs = −0.110, p = 0.534	rs = −0.067, p = 0.400	p = 1.000	
How often do you use CBT and ACT	rs = −0.027, p = 0.782	rs = −0.014, p = 0.940	rs = −0.025, p = 0.773	p = 1.000	
Years using CBT and ACT	rs = −0.027, p = 0.838	rs = 0.125, p = 0.657	rs = −0.006, p = 0.960	p = 1.000	
Familiarity with mindfulness	rs = 0.053, p = 0.558	rs = 0.035, p = 0.844	rs = 0.057, p = 0.474	p = 1.000	
How often do you use mindfulness	rs = 0.147, p = 0.114	rs = −0.021, p = 0.907	rs = 0.106, p = 0.198	p = 1.000	
Years using mindfulness	rs = 0.077, p = 0.485	rs = −0.272, p = 0.179	rs = 0.012, p = 0.900	p = 1.000	
How often do you practice mindfulness?	rs = −0.015, p = 0.903	r~s~ = 0.265, p = 0.245	rs = 0.055, p = 0.606	p = 1.000	
General education	rs = −0.056, p = 0.537	rs = −0.135, p = 0.446	rs = −0.072, p = 0.365	p = 1.000	
Work experience in years	rs = 0.175, p = 0.050	rs = 0.052, p = 0.768	rs = 0.138, p = 0.081	p = 1.000	
Work hours per week	rs = −0.149, p = 0.095	rs = −0.100, p = 0.572	rs = −0.142, p = 0.073	p = 1.000	
How often do you treat chronic pain patients	rs = 0.005, p = 0.957	rs = 0.122, p = 0.491	rs = 0.031, p = 0.698	p = 1.000	
WHOQOL Physical domain	rs = −0.131, p = 0.142	rs = −0.233, p = 0.185	rs = −0151, p = 0.057	p = 1.000	
WHOQOL Psychological domain	r s = −0.204, p = 0.022	rs = −0.031, p = 0.860	r s = −0.173, p = 0.028	p = 0.462	
WHOQOL Social domain	r s = −0.223, p = 0.012	rs = 0.034, p = 0.848	r s = −0.185, p = 0.019	p = 0.252	
WHOQOL Environmental domain	r s = −0.196, p = 0.028	rs = −0.136, p = 0.442	r s = −0.187, p = 0.018	p = 0.378	
WHOQOL Total	r s = −0.259, p = 0.003	rs = −0.134, p = 0.449	r s = −0.235, p = 0.003	p = 0.063	
Notes:

HCPs+CP, Healthcare professionals with chronic pain group; CBT, Cognitive Behavioural Therapy; ACT, Acceptance Commitment Therapy; WHOQOL, World health organizations’ quality of life questionnaire; NPRS, Numerical Pain Rating Scale.

*Bold writing indicates statistically significant findings.

Comparison and exploration of HCPs+CP to H-HCPs (hypothesis 2 and 3)

Since there were no significant differences between the primary and secondary pain groups, we treated both as one group (HCPs+CP, n = 231) for the comparisons with the H-HCPs group (n = 319). Initial comparison of the demographics between the two groups indicated the following significant differences: There were significantly more women and fewer men in the HCPs+CP group (χ2 = 11.268, p = 0.001), the proportions of the professions significantly differed between the groups (χ2 = 35.049, p < 0.001) and body mass index (BMI) was higher in the HCPs+CP group (t = 2.72, p = 0.007) (Tables 5 and 6). The total score of WHOQOL was significantly lower in the HCP+CP than in the H-HCPs (t = −2.136, p = 0.035). We performed another analysis where we removed all the respondents who declared they were not familiar with the intervention and did not use any of the interventions from either group. Following this tapering, the t-test for the WHOQOL indicated no significant differences between the groups in all four domains (physical, social, psychological, and environmental) (Tables 5 and 6). Effect sizes for all WHOQOL domain comparisons between groups were negligible (−0.09 < Cohen’s d < 0.14), suggesting that the practical differences in quality of life between educated HCPs+CP and H-HCPs were minimal. There were no significant differences in all other variables. The pain areas of the HCP+CP are available in Table 7.

Table 5 Ordinal and nominal demographics of the pooled healthcare professionals experiencing pain and healthy healthcare professionals*.

Variables	HCPs+Pain
(n = 231)	H-HCPs
(n = 319)	Significance	
Profession	Bodyworker = 1.7%

Chiropractor = 5.1%

Fitness Instructor/Trainer = 1.2%

Massage therapist = 32%

Medical doctor = 1.2%

Nurse = 4.7%

Occupational Therapist = 2.5%

Osteopath = 7%

Physical Therapist = 35%

Psychologist = 1.2%

Other = 8.4%

	Bodyworker = 2.1%

Chiropractor = 8.7%

Fitness Instructor/Trainer = 1.5%

Massage Therapist = 15.9%

Medical Doctor = 4%

Nurse = 1.5%

Occupational Therapist = 2.2%

Osteopath = 6.5%

Physical therapist = 50.7%

Psychologist = 1%

Other = 5.9%

	χ 2 = 35.049, p = 0.000	
Physical activity	Not at all = 3%

A little (once a week) = 14%

Moderately (2–3 times a week) = 40%

Very much (3–4 times a week = 27%

Extremely (5–7 times a week) = 16%

	Not at all = 6%

A little (once a week) = 9%

Moderately (2–3 times a week) = 39%

Very much (3–4 times a week = 28%

Extremely (5–7 times a week) = 18%

	χ2 = 6.783, p = 0.148	
Alcohol consumption	Not at all = 28%

A little (once per month) = 32%

Moderately (1 time per week) = 23%

Very much (2–3 times a week)= 14%

Extremely (4–7 times a week) = 3%

	Not at all = 25%

A little (once per month) = 28%

Moderately (1 time per week) = 29%

Very much (2–3 times a week) = 14%

Extremely (4–7 times a week) = 4%

	χ2 = 3.597, p = 0.463	
Education	Less than 12 years = 2.5%

High school diploma = 11%

Academic bachelors = 44.5%

Academic Masters = 28.1%

Academic > Masters = 13.9%

	Less than 12 years = 1.5%

High school diploma = 6%

Academic bachelors = 47.6%

Academic Masters = 29.7%

Academic > Masters = 15.2%

	χ2 = 5.910, p = 0.206	
Sector	Public = 33.7%

Private = 50.2%

Both = 16.1%

	Public = 30%

Private = 49.5%

Both = 20.5%

	χ2 = 1.956, p = 0.376	
How often you treat patients with chronic pain	Never = 1.7%

Sometimes = 16%

Very often = 48.5%

Everyday = 33.8%

	Never = 1.5%

Sometimes = 21%

Very often = 40.5%

Everyday = 37%

	χ2 = 4.154, p = 0.245	
Familiarity with PNE	Not at all = 13.5%

A little (once per month) = 18.1%

Moderately (1 time per week) = 32%

Very much (2-3 times a week) = 20.7%

Extremely (4–7 times a week) = 15.7%

	Not at all = 12%

A little (once per month) = 15.6%

Moderately (1 time per week) = 34.1%

Very much (2–3 times a week) = 22.3%

Extremely (4–7 times a week)=16%

	χ2 = 1.079, p = 0.898	
How often you use PNE?	Not at all = 8.5%

On rare occasions = 17.5%

A moderate amount = 32%

Very much = 21%

On regular basis = 21%

	Not at all = 4.6%

On rare occasions = 13.5%

A moderate amount = 35.2%

Very much = 22.5%

On regular basis = 24.2%

	χ2 = 5.019, p = 0.285	
Familiarity with CBT, ACT	Not at all = 10.8%

A little = 21.6%

Moderately familiar = 39.3%

Very familiar = 22%

Extremely familiar = 6.3%

	Not at all = 8.1%

A little = 24.7%

Moderately familiar = 41.3%

Very familiar = 18.1%

Extremely familiar = 7.8%

	χ2 = 3.159, p = 0.532	
How often you use CBT, ACT	Not at all = 37.3%

On rare occasions = 29.1%

A moderate amount = 15.5%

Very much = 10.6%

On regular basis = 7.2%

	Not at all = 36.8%

On rare occasions = 22.5%

A moderate amount = 20.1%

Very much = 10.5%

On regular basis = 10.1%

	χ2 = 4.441, p = 0.350	
Familiarity with mindfulness	Not at all = 6.4%

A little = 21.6%

Moderately familiar = 32%

Very familiar = 26.8%

Extremely familiar = 13.2%

	Not at all = 7.5%

A little = 21%

Moderately familiar = 36.3%

Very familiar = 24.1%

Extremely familiar = 11.1%

	χ2 = 1.801, p = 0.772	
How often you use mindfulness	Not at all = 25.9%

On rare occasions = 23.3%

A moderate amount = 21.6%

Very much = 13.4%

On regular basis = 15.8%

	Not at all = 29.1%

On rare occasions = 29.1%

A moderate amount = 19.4%

Very much = 11.9%

On regular basis = 10.5%

	χ2 = 1.837, p = 0.871	
Do you practice mindfulness yourself?	Yes = 58%

No = 42%

	Yes = 53%

No = 47%

	χ2 = 1.570, p = 0.210	
How often do you practice mindfulness	Once a week = 32%

2–3 times a week = 24.6%

4–5 times a week = 12.6%

Everyday = 30.8%

	Once a week = 33.3%

2–3 times a week = 29.1%

4–5 times a week = 6.2%

Everyday = 31.4%

	χ2 = 3.647, p = 0.302	
Note:

HCP+CP, Healthcare professionals with Chronic Pain Group; H-HCPs, Healthy Healthcare Professionals Group; PNE, Pain Neuroscience Education; CBT, Cognitive Behavioural Therapy; ACT, Acceptance Commitment Therapy. *Bold writing indicates statistically significant findings.

Table 6 Continous demographics of healthy healthcare professionals and the pooled healthcare pain groups (PrimaryPG+SecondaryPG).

Variables	HCPs+CP
(n = 231)	H-HCPs
(n = 319)	Significance	
Gender			χ2 = 9.108, p = 0.003	
Men (%)	68 (30%)	134 (42%)	
Women (%)	163 (70%)	185 (58%)	
Age	43.31 ± 11.08	42.84 ± 11.11	t = 0.495, p = 0.621	
Weight (kg)	76.94 ± 18.01	74.63 ± 16.31	t = 1.556, p = 0.120	
Height (cm)	169.85 ± 10.82	170.69 ± 10.87	t = −0.919, p = 0.359	
BMI	26.71 ± 5.99	25.48 ± 4.48	t = 2.72, p = 0.007	
Average pain intensity in the past week (NPRS)	4.10 ± 1.77	NA	NA	
Years of living with pain	8.63 ± 9.7	NA	NA	
Number of smokers (%)	n = 12 (5%)	n = 13 (4%)	χ2 = 0.387, p = 0.534	
Number of cigarettes per day	8 ± 8.85	4.88 ± 4.36	t = 1.259, p = 0.218	
Clinical experience (years)	15.59 ± 10.72	14.54 ± 10.97	t = 1.117, p = 0.264	
Weekly work hours	32.88 ± 12.75	33.57 ± 12.58	t = −0.655, p = 0.513	
Years of using PNE	6.91 ± 6.85	6.15 ± 5.67	t = 1.294, p = 0.196	
Years of using CBT, ACT	5.07 ± 5.231	5.01 ± 5.50	t = 0.098, p = 0.922	
Years using mindfulness	6.04 ± 6.43	5.81 ± 6.43	t = 0.335, p = 0.738	
Weekly work hours	32.73 ± 12.90	33.50 ± 12.33	t = −0.703, p = 0.482	
*WHOQOL
Physical domain	13.46 ± 1.46	13.26 ± 1.64	t = 1.034, p = 0.302
Cohen’s d = 0.13	
*WHOQOL
Psychological domain	14.22 ± 1.76	14.20 ± 1.99	t = 0.082, p = 0.935
Cohen’s d = 0.01	
*WHOQOL
Social domain	14.42 ± 3.19	14.72 ± 3.171	t = −0.791, p = 0.430
Cohen’s d = −0.09	
*WHOQOL
Environmental domain	15.87 ± 2.20	15.55 ± 2.33	t = −0.783, p = 0.434
Cohen’s d = 0.14	
*WHOQOL
Total score	57.98 ± 6.67	57.76 ± 7.61	t = 0.259, p = 0.795
Cohen’s d = 0.03	
Notes:

HCPs+CP, Healthcare professionals with Chronic Pain Group; H-HCPs, Healthy Healthcare Professionals Group; PrimaryPG, Primary Pain Group; SecondaryPG, Secondary Pain Group; PNE, Pain Neuroscience Education; BMI, Body Mass Index; NPRS, Numerical Pain Raing Scale; CBT, Cognitive Behavioural Therapy; ACT, Acceptance Commitment Therapy; WHOQOL, World Health Organization’s Quality of Life Questionnaire.

* The results of the WHOQOL are presented after removing all the HCPs who declared they have no familiarity nor use any of the approaches. In the HCPs+CP there were n = 117 and in the H-HCPs n = 172.

Table 7 Proportions of the pain areas in the healthcare professional experiencing chronic pain.

HCPs+CP group (n = 231)	
Pain area	Proportion	
Head	9.1%	
Face, jaw, mouth, teeth	6.9%	
Neck	23.4%	
Upper back	19.9%	
Shoulder	18.6%	
Elbow	3.9%	
Wrist, hand, and fingers	17.7%	
Lower back	35.9%	
Stomach	6.9%	
Pelvic	5.6%	
Genitals	1.7%	
Hips and thighs	18.2%	
Knees and calves	16.9%	
Ankles, feet, and toes	13%	
Notes:

HCPs+CP, Healthcare professionals with chronic pain group.

Respondents could pick as many pain areas as they like.

There were no significant differences in the proportions of pain areas between the primary pain group and the secondary pain group (0.001 < χ2 < 2.432; 0.119 < p < 0.988).

When both pain groups were pooled together (HCPs+CP), the same correlations of pain intensity with all other independent variables (education/familiarity, experience, and WHOQOL) remained significant for WHOQOL psychological, social, environmental domains, and its total score (−0.235 < rs < −0.173, 0.003 < p < 0.028). However, after Bonferroni correction, none remained significant (See Table 5).

Assessing perception of H-HCPs with resolved chronic pain (Hypothesis 4)

Concerning hypothesis 4, a total of 146 from the H-HCPs group answered that they had had chronic pain that disappeared. The proportion of HCPs in this group who were unfamiliar with PNE was 28.1%, for CBT and ACT 33.5% and 28.1% for mindfulness. The proportion of H-HCPs familiar with the interventions but not using them was 22.3% for PNE, 65% for CBT and ACT, and 55.5% for mindfulness. The mean years of experience for H-HCPs in this group were PNE = 5.58 ± 4.98, CBT and ACT = 3.92 ± 4.02, and mindfulness = 5.83 ± 5.97. Some 70 H-HCPs said they practice mindfulness more than once a week. Finally, the perceptions of the H-HCPs who have had chronic pain that disappeared revealed that only 11% (n = 16/146) of them perceived that educational or cognitive (psychological) interventions led to their recovery from chronic pain (Table 8).

Table 8 Proportions of the perceptions of healthy healthcare professionals who recovered from chronic pain.

Respondents were first asked: “Have you had chronic pain in the past that has disappeared spontaneously or due to an intervention?” If they answered Yes, they were given the following question:		
“What intervention resolved your pain (You may choose up to 3 answers)?”
n = 146/319		
Perception	N and Proportion	
Movement/exercise	n = 61, 41.8%	
Physical Therapy/rehabilitation	n = 32, 21.9%	
Nothing-Spontaneous recovery	n = 27, 18.5%	
Educational or cognitive (psychological) based approaches	n = 16, 11%	
Alternative medicine	n = 15, 10.3%	
Pharmaceutical	n = 8, 5.5%	
Surgical	n = 3, 2.1%	
Other	n = 22, 15.1%	

Discussion

This exploratory study investigated whether HCPs experiencing chronic pain who were knowledgeable about PNE and cognitive-based interventions for pain derived benefit therefrom by achieving reduced pain intensity and better quality of life. The results did not support those premises and led us to reject hypotheses 1, 2, and 4. In contrast, hypothesis 3 was supported as the HCPs+CP group who were educated in, familiar with, and experienced in using cognitive-based interventions reported quality of life scores equal to healthy HCPs. This finding aligns with other observations suggesting those interventions are associated with lessening disability and improved quality of life in people experiencing chronic pain (Hajihasani et al., 2019; Hilton et al., 2017; Siddall et al., 2022). The findings also indicate that very few HCPs (n = 16) believe pain education and cognitive-based interventions aided their recovery (hypothesis 4).

Given these observations, it is worthwhile to consider how the broader literature has also reported conflicting evidence regarding the efficacy of cognitive-based interventions for chronic pain. Despite the growing popularity of cognitive-based interventions for treating chronic pain states, the scientific evidence of their therapeutic efficacy is unremarkable and contradictory (Cuenca-Martínez et al., 2023; Louw et al., 2021; Martinez-Calderon et al., 2023; O’Connell et al., 2023; Pei et al., 2021; Ram et al., 2023; Sanabria-Mazo et al., 2023; Wood & Hendrick, 2019). For example, a recent umbrella review indicated that it is impossible to make clear clinical recommendations for delivering PNE to chronic pain states based on current meta-analyses (Martinez-Calderon et al., 2023). This conclusion contrasts two recent randomized clinical trials on PNE-like interventions with mindfulness that boasted extraordinary benefits such as complete chronic back pain relief (Ashar et al., 2022; Donnino et al., 2021), whereas mindfulness delivered on its own has unclear effects on pain (Hilton et al., 2017; Sanabria-Mazo et al., 2023). On the other hand, PNE and CBT have demonstrated the ability to deliver benefits unrelated to pain intensity, like decreased disability, anxiety, depression, increased functional capacity, and improved quality of life (Hajihasani et al., 2019; Sanabria-Mazo et al., 2023; Wood & Hendrick, 2019).

Our findings directly challenge premises 1 and 2 of our formal logical argument, which, although simplified, represents the prevailing logic in pain research and clinical practice (Hilton et al., 2017; Louw et al., 2021; Sanabria-Mazo et al., 2023; Tlach & Hampel, 2011; Wetherell et al., 2011). People experiencing chronic pain face a complex interplay of biological, psychological, and social factors that influence their experience, including genetic predispositions, tissue damage, inflammatory processes, psychological states, and socioeconomic conditions (Dunn et al., 2024). Despite this complexity, the field continues to employ the simplistic logic that increased knowledge and cognitive engagement with pain concepts will directly translate to improved pain outcomes in both research and in practice (Louw et al., 2021). The main observation from this study is that HCPs, despite being reasonably educated about pain in general and, more specifically, about PNE and cognitive-based approaches, continue to experience chronic pain. This finding challenges the core assumptions and the logic underlying educational and cognitive-based interventions for pain management that we clearly and formally outlined in Box 1. Theoretically, a comprehensive understanding of pain and its relevant mechanisms is expected to diminish fear and enhance the accuracy of pain-related predictions, thereby facilitating pain relief (Pei et al., 2021). However, the persistent pain experienced by these educated HCPs raises substantial doubts about the effectiveness of these strategies. If such knowledge does not ameliorate the pain experienced by the practitioners themselves, it casts skepticism on the potential for patients to achieve better outcomes through similar educational interventions. This discrepancy prompts a reevaluation of the foundational principles of educational and cognitive approaches to pain management.

When considering the mentioned theoretical frameworks of cognitive-based interventions, this study’s findings indicate that mere education or familiarity with these approaches does not significantly impact pain intensity or alter the pain trajectory among HCPs. A recent meta-analysis supports this assertion (Ram et al., 2023). Our results and those from other studies reveal that while proficiency in PNE and other cognitive-based interventions may enhance quality of life, it does not modify pain intensity or its progression (Hajihasani et al., 2019; Sanabria-Mazo et al., 2023; Sousa et al., 2024; Wood & Hendrick, 2019). This pattern is reminiscent of the framing effect described by Tversky & Kahneman (1981), which demonstrate that the way information is presented can heavily influence outcomes without altering the underlying content (Tversky & Kahneman, 1981). In other words, the apparent efficacy of those interventions observed in clinical trials may be largely attributed to contextual influences and self-conviction–akin to positive framing-rather than the specific educational content or treatment effects. In the case of PNE, it is plausible that the benefits of any educational intervention may be similarly achieved through positive framing, regardless of the content. Recent studies on PNE support this premise (Adenis et al., 2024; Ponce-Fuentes et al., 2023). This raises critical questions about PNE’s unique value, underscoring the need for comparative studies that evaluate its effectiveness against other positively framed or sham educational interventions to validate these findings further. However, we are unaware of such studies.

While the current study did not find a reduction in pain intensity among HCPs knowledgeable about cognitive-based interventions, these individuals reported quality-of-life metrics comparable to those of their healthy peers. This indicates that, although these interventions may not decrease pain intensity, they do enhance life satisfaction by improving emotional resilience and psychological well-being. This finding aligns with previous research on cognitive-based interventions, underscoring their potential to improve quality of life (Hajihasani et al., 2019; Sanabria-Mazo et al., 2023; Wood & Hendrick, 2019). Future research should investigate which components of these interventions are most effective at enhancing the quality of life, considering that the educational content and level alone do not contribute to the treatment effect.

Unlike PNE, which primarily reframes pain through education, ACT, CBT, and mindfulness actively engage patients in coping with their predicaments. ACT is said to promote psychological flexibility by fostering acceptance of pain and commitment to value-based actions (Hayes et al., 2013) whereas CBT presumably targets and restructures maladaptive thought patterns (de C Williams et al., 2020; Fenn & Byrne, 2013; Sanabria-Mazo et al., 2023). Similarly, mindfulness practices are aimed to improve emotional regulation and resilience. The comparable quality-of-life scores observed in our study among HCPs with chronic pain might reflect the beneficial effects of these interventions, which together can enhance overall well-being even in the absence of significant reductions in pain intensity (Hajihasani et al., 2019; Siddall et al., 2022). Future research should further differentiate the unique contributions of ACT, CBT, and mindfulness from educational approaches like PNE in chronic pain management.

A strength of this study is the rigorous bias reduction strategies we employed. We were able to exclude survivorship bias in the H-HCPs group as the majority whose pain had resolved did not attribute this resolution to cognitive or educational interventions (only 11%). Their most common belief was that their pain was resolved through movement/exercise or rehabilitation, with spontaneous recovery being the next most common belief. In line with the previous point of discussion, it is important to mention that physical therapy interventions including exercise/movement, often naturally integrate psychological components such as safety messaging and motivation, making it difficult to completely separate their physical and cognitive elements.

The perception of those HCPs that their pain spontaneously resolved is vital to open a discussion on the natural history of chronic pain conditions; that is a scarcely mentioned topic and not adequately considered by researchers and the general public as the reason for success in clinical settings and clinical trials. For example, Chronic Regional Pain Syndrome type 1 (Sandroni et al., 2003) has been shown to resolve naturally, and both spontaneous remission and relapse have been documented in vulvodynia, where it is believed that persistence without remission is the exception rather than the rule (Reed et al., 2016). Another example is low back pain, which was once defined as acute, subacute, or chronic based on the duration of the present episode. However, low back pain is now understood as an episodic condition (Kongsted et al., 2016). When interventions are applied close to a natural resolution/remission point of these conditions, the natural reflexive conclusion can be that the treatment works. This phenomenon casts doubt upon the many anecdotal reports about people who report recovery from chronic pain due to various interventions and even clinical trials. Consequently, HCPs and researchers should be cautious when attributing success to interventions without considering natural resolution or remission of the conditions.

Several limitations of this study warrant acknowledgment. A potential critique concerns our classification criteria for primary pain, which may have been overly inclusive and potentially incorporated cases attributable to secondary pain conditions. However, this concern is mitigated by two key factors. Firstly, the majority of reported pain sites in our sample were lower back (35.9%), neck (23.4%), shoulder (18.9%), and upper back (19.9%) (see Table 4), regions frequently associated with non-specific pain conditions in clinical practice and research literature (i.e., primary pain) (Chiarotto & Koes, 2022; Kazeminasab et al., 2022; Park et al., 2020; Risetti et al., 2023). Secondly, our study population consisted of educated HCPs who, given their professional training and knowledge, can likely distinguish between specific and non-specific pain etiologies. Their self-reported pain classifications are presumably more accurate than the general population’s. An additional limitation is our decision to collapse the primary and secondary pain groups for subsequent analyses after finding no significant differences between them. While this approach was statistically justified based on our initial comparisons, it potentially obscures meaningful clinical distinctions between these pain categories that might emerge with larger samples or more nuanced assessment methods. By combining these groups, we may have diluted potential subgroup effects specific to either primary or secondary pain mechanisms.

Another potential critique of our methodology is the absence of a formal assessment of the respondents’ educational quality, particularly regarding their understanding of pain neuroscience. While tools such as the Neurophysiology of Pain Questionnaire could have been employed to this end, we contend that such an approach would have been both unnecessary and potentially counterproductive in the context of our study. Firstly, including additional assessment tools would have significantly increased the survey’s length and complexity, potentially compromising completion rates and, by extension, the robustness of our dataset. The balance between comprehensive assessment and participant engagement is delicate, and we prioritized maintaining a high response rate to ensure a representative sample (Rolstad, Adler & Rydén, 2011).

Moreover, our study population is comprised of educated HCPs whose baseline knowledge of pain science and cognitive-based approaches can already be reasonably assumed to exceed that of the general population. Their professional credentials and ongoing clinical practice serve as a proxy for their cognitive engagement with pain concepts, validating a key premise of our research without necessitating further testing. What further supports our argument is that our study population consisted primarily of physiotherapists (44%) from anglophone countries with established pain education curricula (Canada, United States of America, United Kingdom, and Australia proportion = 50.9%) and evidence which consistently demonstrates that they receive more extensive training in pain mechanisms compared to other disciplines (Briggs, Carr & Whittaker, 2011; Hoeger Bement & Sluka, 2015; Mezei, Murinson & Team, 2011; Shipton et al., 2018; Watt-Watson et al., 2009).

Furthermore, it is worth noting that cognitive approaches to pain management, despite their theoretical complexity, often distill into relatively straightforward, accessible messages when applied clinically. The proliferation of media and educational resources, apps, and tools aimed at simplifying these concepts for laypersons further supports the notion that a detailed assessment of our respondents’ grasp of cognitive-based approaches might yield diminishing returns regarding insights gained (Choi, 2021; Devan et al., 2019; Sage et al., 2008; Weisman et al., 2022).

In light of these considerations, we believe our approach strikes an appropriate balance between methodological rigor and practical constraints, allowing us to focus on our primary research questions without unduly burdening participants or introducing potential confounders. Future studies might explicitly explore the relationship between detailed cognitive approaches, knowledge, and clinical outcomes. Still, for the purposes of our investigation, the participants’ professional status and self-reported familiarity with cognitive-based interventions provided sufficient context for our analyses.

Although online dissemination methods are valid for reaching the desired population (Weisman et al., 2021; Yona et al., 2022), selection bias cannot be wholly excluded. Finally, while the desired sample size was reached, many respondents did not complete the questionnaire, and not all health professions and countries were equally represented. An increased completion rate would have allowed for more subgroup analyses and exploration of the effects of countries (i.e., culture), which may have yielded different quality-of-life scores. Finally, other factors that could have influenced our findings were not assessed, including sociodemographic and socioeconomic profiles, sleep quality, night shift work, and the extent of direct work with chronic pain patients.

Conclusion

HCPs who live with chronic pain, are familiar with PNE, and are experienced in cognitive-based interventions such as CBT, ACT, and mindfulness do not correlate with reported lowered pain intensity. However, the quality of life of those HCPs was similar to that of healthy HCPs. Very few healthy HCPs whose chronic pain had resolved believe that education about pain or cognitive-based interventions contributed to this resolution. Most HCPs with resolved pain believed that movement/exercise and physical therapy contributed to their pain resolution or that their pain resolved spontaneously. The findings question the theoretical basis of those interventions, namely those that apply the biopsychosocial model or the reasoning of predictive processing to assess for causality of pain and their use as self-management tools.

Supplemental Information

Supplemental Information 1 Raw Data.

Supplemental Information 2 STROBE Statement.

Additional Information and Declarations

Competing Interests

The authors declare that they have no competing interests.

Author Contributions

Asaf Weisman conceived and designed the experiments, performed the experiments, analyzed the data, prepared figures and/or tables, authored or reviewed drafts of the article, and approved the final draft.

Tomer Yona conceived and designed the experiments, performed the experiments, analyzed the data, prepared figures and/or tables, authored or reviewed drafts of the article, and approved the final draft.

Youssef Masharawi conceived and designed the experiments, performed the experiments, analyzed the data, prepared figures and/or tables, authored or reviewed drafts of the article, and approved the final draft.

Human Ethics

The following information was supplied relating to ethical approvals (i.e., approving body and any reference numbers):

Tel-Aviv University ERB.

Data Availability

The following information was supplied regarding data availability:

Raw data.

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
