# Peer review of "Education about pain and experience with cognitive-based interventions do not reduce healthcare professionals’ chronic pain"

_PeerJ, doi:10.7717/peerj.19448_

## Round 0.1 · original submission · Major Revisions

Three reviewers have provided their comments. Please respond to them in full

Reviewer 1 ·

Basic reporting

Please see attached document.

Experimental design

Please see attached document.

Validity of the findings

Please see attached document.

Additional comments

Please see attached document.

Annotated reviews are not available for download in order to protect the identity of reviewers who chose to remain anonymous.

·

Basic reporting

No comment. All the content is contextually coherent and strongly relevant.

Experimental design

No comment.

Validity of the findings

No comment.

Additional comments

The introduction section discloses contradictory ideas as if the discussion section, hence undermines the brief recalling related to CBT, PNE, etc. Those should be adverted in short terms for the readers who are not closely familiar with the jargon. Furthermore, contradictory ideas might be migrated to the discussion section. However, the whole idea leaves no gaps to intervene.

·

Basic reporting

The authors have met the basic reporting criteria, but the manuscript would benefit from a deeper exploration of the context of cognitive therapy delivery.

Introduction
The authors refer to the amount of pain education provided for various healthcare professionals, but their argument to pick the healthcare professionals who have been given the most falls down as they have not considered vet practitioners who in fact have far and away a greater number of hours of pain education. One would need to look at their curriculum to ascertain whether any of those hours are dedicated to cognitive strategies (perhaps not), but perhaps the authors could consider this aspect and make any adjustments they feel as necessary?

A body of literature that has been missed in the justification in the introduction is that by GL Moseley who has been a leader in the field of pne. Some useful retrospectives on the subject might inform a deeper understanding of the aim of pne and what it comprises as this is often missed. Teaching Patients About Pain: The Emergence of Pain Science Education, its Learning Frameworks and Delivery Strategies
Lorimer Moseley, G. et al. The Journal of Pain, Volume 25, Issue 5, 104425

Experimental design

There are three main concerns with the logic argument:
1. The premise that cognitive based interventions alone will aid recovery from chronic pain. It is widely accepted that chronic pain is influenced by many factors and some of those are difficult to shift at different times in one’s life (financial stress, family illness, a poor decision which has impacted life) no matter how well or skillfully you apply cognitive strategies, other things might also need to line up for change to happen. The authors did not collect sociodemographics such as socioeconomic status/household finances/living conditions which may have shed some light on this. This challenges premise 2 in the logic argument and hypothesis 4. The information collected by the authors in this study does not attempt to address that and hence there is limited interpretation that can be made from the outcomes collected in the study. Would the authors please consider addressing this point?
2. Premise 3 states that HCPs are the most educated about pain, but could the authors please check that this is true by checking the pain education received by veterinarians?
3. The conclusion is challenged by the suggestions in point 1 above and needs to be re-considered
There are two main concerns with the study design
1. The authors have included people with secondary chronic pain on the basis of their answers on the survey that report they have had surgery/cancer or some other primary disorder from which the pain is secondary. But at no stage has the primary disorder been linked to the pain, and this assumption (‘presumed’ as the authors stated) makes the interpretation of outcomes for secondary pain spurious at best. Could the authors please justify this feature of the study design? This assumption affects the set-up of hypothesis 1.
2. The authors collapsed both pain groups in the end anyway, please consider the limitations of doing this and add to the limitations section.
3. The authors asked “what intervention resolved your pain (you may chose up to 3 answers)” when respondents answered yes to having ever had chronic pain that has disappeared spontaneously or due to an intervention and presented alternatives. When the authors reported on these outcomes the word ‘resolved’ became ‘recovered’. There are subtle differences with the words – resolved means to cognitively solve a problem and recovered implying some active effort on the person’s part to get back to their ‘normal’. They are not synonymous. Could the authors please reconsider their use of these terms in reporting results and discussion.

Validity of the findings

Results
Demographics were collected data such as smoking, years of mindfulness and quality of life, yet the authors reported these in a table, but only for the healthcare professionals with pain. Could the authors please explain why the HC without pain were not included in the table?

In table 3 it can be seen that the percentage of massage therapists experiencing any chronic pain is similar to that of the physical therapists. How many hours do Massage therapists have regarding pain theory and practice in their courses? Is it possible that the strenuous nature of both physical therapy and massage therapy make them vulnerable to chronic pain states? Could the authors compare outcomes for this HCP as a sensitivity analysis?

Discussion
The discussion could integrate a deeper understanding of the factors influencing pain education and cognitive therapies.

Conclusion
The conclusion states that most of the recovered HCPs believed that movement/exercise and physical therapy contributed to their recovery – given the rationale stated by the authors that physios have more pain education – don’t the authors think that movement/exercise provided by physical therapy might be psychologically informed and in fact the component that are exercise would be difficult to separate from the psychological components that incorporate safety messages and motivation/credibility statements?

Additional comments

This is a novel area of investigation.

---

## Round 0.2 · accepted · Accept

Dear Authors,

I would like to express my appreciation for your patience and efforts to improve the quality of the manuscript. Two experts have now endorsed your submission for acceptance of publication in PeerJ. Congratulation!!!

Thank you for submitting your article to PeerJ. I look forward to receiving your research and review articles in the future.

Best Regards
Ph.D. Yung-Sheng Chen

·

Basic reporting

No comment

Experimental design

No comment

Validity of the findings

No comment

Additional comments

No comments.

Reviewer 4 ·

Basic reporting

Congratulations to the authors. The manuscript has improved considerably. I have no further comments.

Experimental design

No comments.

Validity of the findings

No comments.

Additional comments

No comments.